# Non-volatile electrically programmable integrated photonics with a 5-bit operation

Rui Chen [1] ✉, Zhuoran Fang[1], Christopher Perez[2], Forrest Miller[1], Khushboo Kumari[1], Abhi Saxena[1], Jiajiu Zheng [1], Sarah J. Geiger[3], Kenneth E. Goodson[2] & Arka Majumdar [1,4] ✉

Scalable programmable photonic integrated circuits (PICs) can potentially transform the current state of classical and quantum optical information processing. However, traditional means of programming, including thermo-optic, free carrier dispersion, and Pockels effect result in either large device footprints or high static energy consumptions, significantly limiting their scalability. While chalcogenide-based non-volatile phase-change materials (PCMs) could mitigate these problems thanks to their strong index modulation and zero static power consumption, they often suffer from large absorptive loss, low cyclability, and lack of multilevel operation. Here, we report a wide-bandgap PCM antimony sulfide ($Sb_2S_3$)-clad silicon photonic platform simultaneously achieving low loss (<1.0 dB), high extinction ratio (>10 dB), high cyclability (>1600 switching events), and 5-bit operation. These $Sb_2S_3$-based devices are programmed via on-chip silicon PIN diode heaters within sub-ms timescale, with a programming energy density of $\sim 10\,fJ/nm^3$. Remarkably, $Sb_2S_3$ is programmed into fine intermediate states by applying multiple identical pulses, providing controllable multilevel operations. Through dynamic pulse control, we achieve 5-bit (32 levels) operations, rendering $0.50 \pm 0.16$ dB per step. Using this multilevel behavior, we further trim random phase error in a balanced Mach-Zehnder interferometer.

Programmable photonic integrated circuits (PICs), usually composed of arrays of tunable beam splitters and phase shifters, can change their functionalities on demand. This flexibility has recently extended their traditional applications from optical interconnects to optical computing[1], optical programmable gate arrays[2], and quantum information processing[3]. Further scaling of the programmable PICs requires the constituent devices to have a smaller footprint and lower power consumption. However, current programmable PICs are primarily based on weak tuning mechanisms, such as thermo-optic effects[4,5], free-carrier effects[6], and Pockels effects[7], which are traditionally optimized for low switching power and higher speed application. Unfortunately, they provide a small change in the refractive index (Δn < 0.01),

resulting in a large footprint (> 100 μm). Plasmonic effects can shrink the device footprint to ~10 μm[8,9], but often result in lossy devices, precluding usage in large-scale systems. Thermo-optic effects incur severe thermal crosstalk, necessitating additional heaters and control circuits for crosstalk compensation. Most importantly, these effects are all volatile and require a constant power supply (~10 mW[1]). Such high static power consumption is prohibitive for programmable PICs, which generally need infrequent programming;[10] example applications include post-fabrication trimming[11], setting weights in a photonic neural network for fast inference[1,12,13], and connecting functioning devices, including quantum emitters on-chip[14]. We emphasize that, the traditional metrics like switching speed and switching energy for these

[1]Department of Electrical and Computer Engineering, University of Washington, Seattle, WA 98195, USA. [2]Department of Mechanical Engineering, Stanford University, Stanford, CA 94305, USA. [3]The Charles Stark Draper Laboratory, Cambridge, MA 02139, USA. [4]Department of Physics, University of Washington, Seattle, WA 98195, USA. ✉e-mail: charey@uw.edu; arka@uw.edu

applications are of little importance, as the switching events occur very infrequently.

Chalcogenide-based non-volatile phase-change materials (PCMs) can provide a unique solution for these applications[15–17]. PCMs have two stable, reversibly switchable micro-structural phases (amorphous and crystalline, termed here as a- and c-phase), with drastically different optical refractive indices ($\Delta n \sim 1$). Thanks to the non-volatile phase transition, no static power is needed to hold the state upon switching PCMs' phase. The substantial refractive index change $\Delta n$ and nonvolatility enable compact reconfigurable devices (~10 $\mu m^{10,18}$) with zero static energy consumption[19–24]. Moreover, PCMs are compatible with large-scale integration since they can be easily deposited by sputtering[19,21,22,24,25] or thermal evaporation[18] onto almost any PIC materials, including silicon and silicon nitride. Despite these advantages, archetypal PCMs such as $Ge_2Sb_2Te_5$ (GST) and GeTe exhibit strong absorption in both phases at relevant optical communication wavelengths. This hinders their utility in phase shifters – an essential building block for programmable PICs. Emerging wide-bandgap PCMs, such as GeSbSeTe (GSST)[26], antimony selenide ($Sb_2Se_3$)[27], and antimony sulfide ($Sb_2S_3$)[28] can circumvent this loss and have recently generated strong interest in the community[18,23,29–31]. In particular, $Sb_2S_3$ shows the widest bandgap among all these PCMs, allowing transparency down to ~600 nm in the amorphous phase[28]. Moreover, the lack of selenium in $Sb_2S_3$ makes it less toxic[32] and less likely to contaminate the chamber during sputtering or evaporation processes. Thus, $Sb_2S_3$ is much more amenable to be adapted in a commercial foundry. Despite these promises, previous works mostly switched a $Sb_2S_3$ blanket film optically[28,33] or a $Sb_2S_3$ metasurface[30,31], usually with sophisticated femto-second pulse lasers. Very few electrical controls of $Sb_2S_3$ were experimentally demonstrated[21,28] and high-endurance electrical control of $Sb_2S_3$ remained unsolved. Such electrical actuation capabilities could enable large-scale PCM-PIC systems, where a large number of devices are simultaneously controlled, far beyond the reach of the all-optical scheme.

Here, we demonstrate electrically controlled $Sb_2S_3$-clad silicon photonic devices with low insertion loss (<1 dB), high extinction ratio (>10 dB), and high endurance (>1600 switching events). The phase transition is actuated by silicon PIN (p-doped-intrinsic-n-doped) diode heaters. We established the versatility of this hybrid platform with three different integrated photonic devices, including microring resonators, Mach-Zehnder interferometers (MZIs), and asymmetric directional couplers. We also observed multilevel (32 levels) operation, highest among all reported electrically controlled PCM-silicon platforms, with a resolution of $0.50 \pm 0.16$ dB per step. This multilevel operation is achieved by sending multiple thermally separated (~1 s) near-identical electrical pulses for both amorphization and crystallization. We leveraged this multilevel behavior to trim a balanced MZI to correct the random phase error caused by fabrication imperfections. The multilevel operation is crucial to avoid under or over-correction during trimming.

## Results

We characterized the sputtered $Sb_2S_3$ thin films to obtain the refractive indices and verify the crystallization capability (annealed under 325 °C in Nitrogen environment for 10 mins) of the as-deposited a-$Sb_2S_3$ (Supplementary Fig. 1). The silicon photonic devices (Fig. 1–3) were designed to operate at the telecommunication O-band (1260–1360 nm) and fabricated on a standard silicon-on-insulator (SOI) wafer with 220 nm silicon and 2 μm buried oxide. The 500-nm-wide waveguides are fabricated by partially etching 120-nm silicon. We then deposited 450-nm-wide $Sb_2S_3$ onto the SOI chip via sputtering. The slightly smaller width than the waveguide is to compensate for electron beam lithography (E-beam) overlay tolerance. Our simulation results show a change in effective index $\Delta n_{eff} \approx 0.018$ between a- and c-$Sb_2S_3$ (Supplementary Fig. 2). The $Sb_2S_3$ films are electrically

controlled via on-chip silicon PIN micro-heaters[22,24]. The $p^{++}$ and $n^{++}$ doping regions were designed 200 nm away from the waveguide to avoid free-carrier absorption loss[24], indicated in the scanning electron microscope (SEM) images with false colors in Fig. 1c, Fig. 2c, and Fig. 3c. The $Sb_2S_3$ stripes are encapsulated with 40 nm of $Al_2O_3$ grown by atomic layer deposition (ALD) under 150 °C. This conformal encapsulation is critical to prevent $Sb_2S_3$ from oxidation and thermal reflowing, and is essential to attain high endurance. To show that our $Sb_2S_3$-clad silicon photonic platform is versatile and compatible with most PIC components, we demonstrate three widely used PIC components: (1) a microring resonator to show low-loss tuning of cavities, (2) a balanced MZI to demonstrate a full π phase shift and a broadband operation, and (3) an asymmetric directional coupler to create a compact programmable unit (see simulation results in Supplementary Fig. 3).

### Nonvolatile microring switch integrated with $Sb_2S_3$ phase shifter

We deposited 10-μm-long, 20-nm-thick $Sb_2S_3$ on a micro-ring resonator with 30 μm radius (Fig. 1a–c). The free spectral range (FSR) is ~2.42 nm (Fig. 1d), and the bus-ring gap is 280 nm to achieve a near-critically coupled device. We switched the as-deposited a-$Sb_2S_3$ to c-$Sb_2S_3$ on the microring resonator by applying three 1.6 V, 200-ms-long pulses with ~3.4 mJ energy (or SET pulses) separated by 1 s and then re-amorphized the material via three 7.5 V, 150-ns-long pulses with ~56 nJ energy also separated by 1 s (RESET pulses). The unit energy density (energy/total $Sb_2S_3$ volume) for switching $Sb_2S_3$ is then estimated as 0.6 $fJ/nm^3$ (38 $pJ/nm^3$) for amorphization (crystallization). We note that the 200-ms-long SET pulse is indeed significantly longer than other reported PCMs, such as GST (50 μs[24] or 100 μs[23]) and $Sb_2Se_3$ (5 μs[18] or 100 μs[23]), and causes a large crystallization energy. But we found that the SET pulse duration can be reduced to around 100 μs after the first few cycles (see Method and Supplementary Fig. 15). Therefore, the crystallization energy is reduced to ~1.7 μJ (19 $fJ/nm^3$) after the initial conditioning, comparable to other PCMs[18,24]. With the 100 μs crystallization pulse condition, this device could be operated at ~ kHz speed for a complete SET/RESET cycle. The slow crystallization, however, can allow amorphizing a large volume of $Sb_2S_3$ (Supplementary Fig. 16). Figure 1d shows a resonance shift of ~0.394 nm upon switching the 10 μm $Sb_2S_3$ from the a- (blue) to the c-phase (orange), corresponding to a π-phase shift length $L_\pi$ of ~30.7 μm, significantly shorter than the 1-millimeter $L_\pi$ of ferroelectric non-volatile phase shifter[34]. The SET and RESET processes were repeated for 10 cycles. The shift in resonance is highly repeatable, as suggested by the slight standard deviation (Fig. 1d). The excess loss from a-$Sb_2S_3$ is negligible[21], and in c-$Sb_2S_3$-clad waveguides the loss is estimated to be 0.024 dB/μm (0.72 dB/π), which is three times larger than our simulation result (0.26 dB/π). We attribute this excess loss to the scattering from the $Sb_2S_3$ thin film due to non-uniform deposition/liftoff and local crystal grains in c-$Sb_2S_3$ (Supplementary Fig. 7). We verified that the loss due to mode mismatch at the transition between the bare silicon waveguide and the $Sb_2S_3$-loaded waveguide is small (~0.013 dB/facet), consistent with the fact that the thin $Sb_2S_3$ film should not significantly change the mode shape (Supplementary Fig. 3).

### Nonvolatile Mach-Zehnder switch integrated with $Sb_2S_3$ phase shifter

Figure 2a–c show a balanced MZI operating at wavelengths between 1320 nm and 1360 nm with both arms covered with 30-μm-long 20-nm-thick $Sb_2S_3$. A multimode interferometer with a 50:50 splitting ratio was designed and fabricated, as shown in Fig. 2d (Simulation in Supplementary Fig. 6). Initially, the light comes out mainly from the bar port with an extinction ratio of ~13 dB (Fig. 2e). The light coming out from the bar port instead of the cross port in this balanced MZI can be explained by the random phase errors in two arms due to fabrication

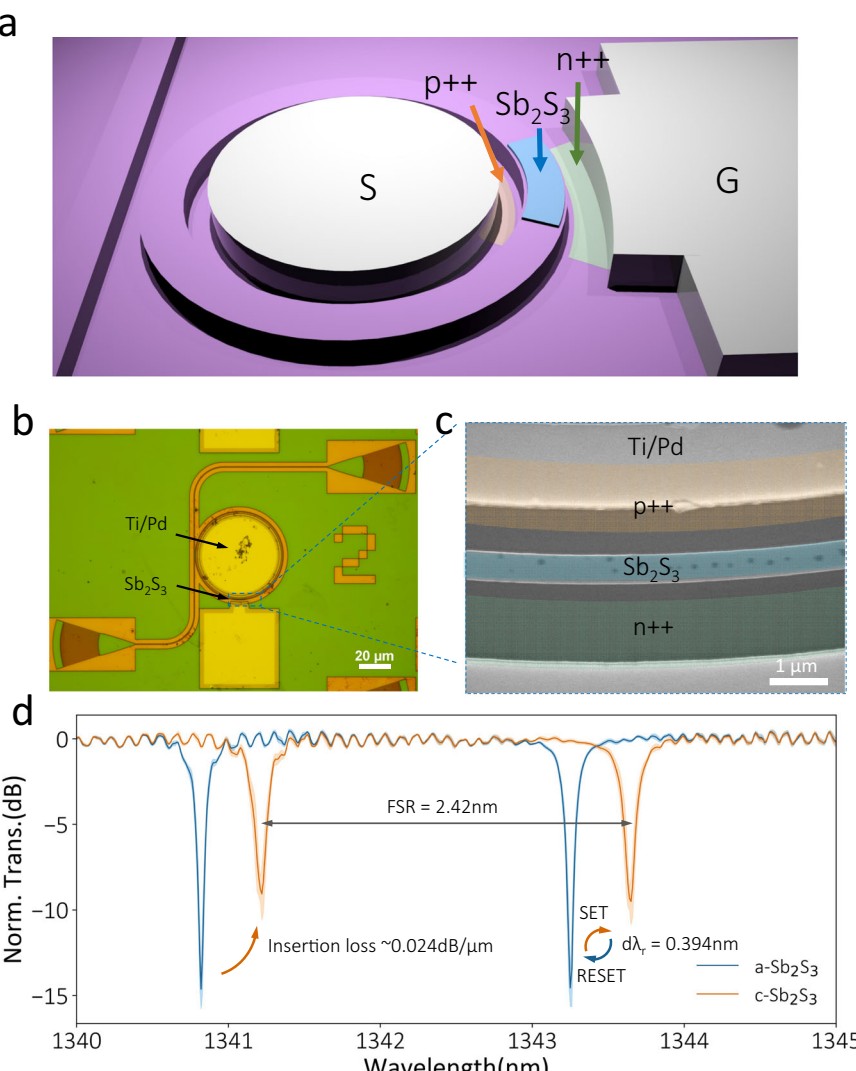

**Fig. 1 | A high-Q ring resonator loaded with 10 μm long 20-nm-thick Sb₂S₃.**
**a** Schematic of the device (the encapsulation ALD Al₂O₃ layer is not shown),
**b** optical, and **c** Scanning-Electron Microscope (SEM) images of the micro-ring
resonator. The Sb₂S₃ thin film, n⁺⁺, and p⁺⁺ doped silicon regions are represented by
false colors (blue, orange, and green, respectively). **d** Measured micro-ring spectra
in two phases (SET: three 1.6 V, 200-ms-long, 3.4 mJ pulses to change Sb₂S₃ to the
crystalline state; RESET: three 7.5 V, 150-ns-long, 56 nJ pulses to change Sb₂S₃ to the
amorphous state). The spectra are averaged over ten cycles of reversible electrical
switching, and the shaded area shows the standard deviation. Norm. Trans. stands
for normalized transmission, normalized to a reference waveguide on the
same chip.

imperfections, especially in the S-bend. One can overcome these fab-
rication imperfections by exploiting a wider waveguide to improve
fabrication robustness[35]. Alternatively, this random phase error can be
corrected using Sb₂S₃ for post-fabrication trimming, as we show later
in this paper. The Sb₂S₃ on one arm was switched by two 1.7 V, 200-ms
SET electric pulses with an energy of 11.6 mJ (density: 42 $pJ/nm^3$) to
provide a full π phase shift. An 8.1 V, 150-ns short RESET electric pulse
with an energy of 197 nJ (density: 0.73 $fJ/nm^3$) switched the device
back to the initial state. Figure 2e, f show the transmission spectra
normalized to a reference waveguide when the Sb₂S₃ film is in the
amorphous and crystalline phases, respectively. The c-Sb₂S₃ displays a
complete spectrum flip, showing a bar state with an extinction ratio of
15 dB. We then recorded the bar port transmission at 1330 nm for
100 switching events without device degradation (Supplemen-
tary Fig. 8).

**Compact asymmetric directional coupler switch**
We also designed and fabricated a compact asymmetric directional
coupler (coupling length $L_c \approx 79$ μm) (simulation in Supplementary

Fig. 3), as shown in Fig. 3a–c. The coupler consists of two waveguides
with different widths. The narrower 409-nm-wide waveguide (hybrid
waveguide) was capped with 20-nm-thick Sb₂S₃ and designed to allow
phase match with the wider 450-nm-wide waveguide (bare waveguide)
for c-Sb₂S₃[36]. As such, the input light could completely couple to the
cross port in one coupling length. Once Sb₂S₃ is switched to the
amorphous state, the effective index of the hybrid waveguide changes
while the bare waveguide remains the same. The resulting phase mis-
match changes the coupling strength and coupling length. Then, co-
optimizing the gap and waveguide length permits a complete bar
transmission. The c-Sb₂S₃ phase matching approach, instead of
a-Sb₂S₃[20,22,36,37], allows a more symmetric performance regardless of
the input port (Supplementary Fig. 4), crucial for a 2 × 2 device. If phase
mismatch happens in the c-Sb₂S₃ state, the slight loss of c-Sb₂S₃ on one
of the waveguides will result in different bar state insertion loss when
the light goes from different input ports. We note that the 79-μm
coupling length can be potentially reduced (~ 34 μm) by depositing a
thicker (50 nm) Sb₂S₃ to provide stronger refractive index modulation
(Supplementary Table 1).

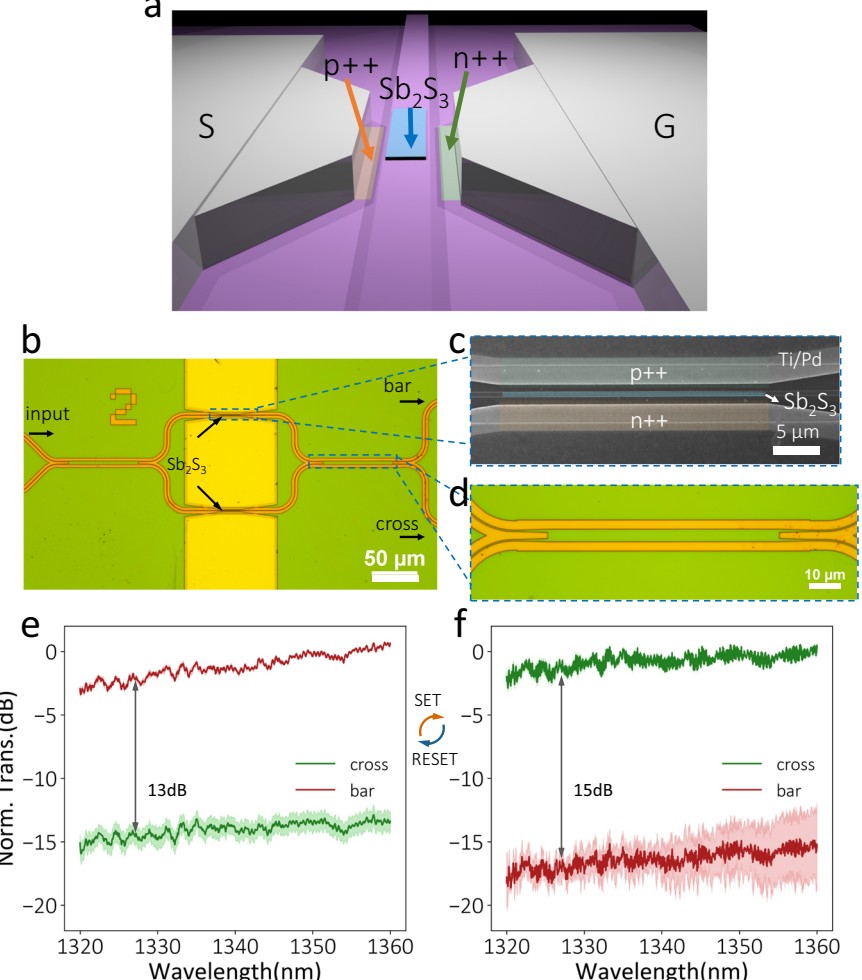

**Fig. 2 | A Mach-Zehnder interferometer with both arms covered with 20-nm-thick Sb₂S₃. a** Sb₂S₃ phase shifter schematic (the encapsulation ALD Al₂O₃ layer is not shown). **b** Optical and (**c**) SEM images of the Sb₂S₃ phase shifter. **d** Optical micrograph of the 50:50 splitting ratio multi-mode interferometer. **e**, **f** Transmission spectra at both bar and cross ports for (**e**) a-Sb₂S₃ (RESET: an 8.1 V, 150 ns, 197 nJ pulse) and (**f**) c-Sb₂S₃ (SET: two 1.7 V, 200 ms, 12 mJ pulses). The green and red lines represent the measured transmission at cross and bar ports. The shaded region indicates the standard deviation of transmission for 5 cycles of measurements.

Figure 3d, e show the transmission spectra for a- and c-Sb₂S₃, switched with three 9.6 V, 500 ns, 922 nJ RESET, and 2.7 V, 200 ms, 29.2 mJ SET pulses, respectively. The energy density for amorphization (crystallization) is 1.28 $fJ/nm^3$ (40 $pJ/nm^3$). The insertion losses are 2 dB (0.5 dB), and the extinction ratios are around 10 dB (11 dB) for a (c)-Sb₂S₃. The unexpected high insertion loss when the Sb₂S₃ is in the amorphous state can be attributed to several factors, including deviation of the gap size from the design, Sb₂S₃ overlay alignment error, or the cross-port grating coupler fabrication imperfection. To estimate the actual loss of the device, we apply the c-Sb₂S₃ loss extracted from the ring resonator to the simulation and calculate this device's insertion loss to be ~0.1 dB (0.9 dB) for a(c)-Sb₂S₃ (Supplementary Fig. 5).

Figure 4 shows >1600 switching events for the asymmetric directional coupler, a demonstration of large-endurance electrical switching of Sb₂S₃. We note that very few electrical controls of Sb₂S₃ were experimentally demonstrated[21,28] and no work has previously shown reversible electrical switching of Sb₂S₃. Limited by our measurement setup, we separately measured the cross (Fig. 4a) and bar ports (Fig. 4b). The higher insertion loss (~1 dB) at around event 500 was due to optical fiber misalignment. We note that almost no performance degradation occurred at the end (Supplementary Fig. 9); hence, 1600 switching events are not the limit of this device.

We stopped the experiment at that point due to the long duration of the experiment. The cross-port transmission shows a relatively large variation for a-Sb₂S₃ (Fig. 4a blue scatterers, from ~−15 dB to ~−35 dB), which was caused by incomplete amorphization or thermal reflow of the Sb₂S₃ film. Since the plot is on a logarithmic scale, such a large variation in a-Sb₂S₃ (due to higher transmission) is not visible in Fig. 4b.

## Multilevel 5-bit operation with dynamic electrical control
Our Sb₂S₃-Si integrated structures further show a stepwise multilevel operation up to 32 levels with dynamic pulse control. In Fig. 5a, we show the multilevel transmission at both cross and bar ports of an asymmetric directional coupler while sending in RESET 10 V, 550 ns, 1.1 μJ pulse every other second to amorphized the Sb₂S₃. We started the experiment with "coarse" tuning, where unoptimized, identical pulses were sent to partially amorphize the c-Sb₂S₃ device to demonstrate multiple levels. The asymmetric directional coupler was originally in the "cross state" (red region). After one partial amorphization pulse, it was reconfigured into an intermediate state (orange region), where light comes out from both cross and bar ports. After six pulses, a complete "bar state" (green region) was achieved. We repeated this experiment five times for each port and plotted the average transmission levels and the standard deviation. The variation is attributed to the stochastic phase change process using electrical controls[38].

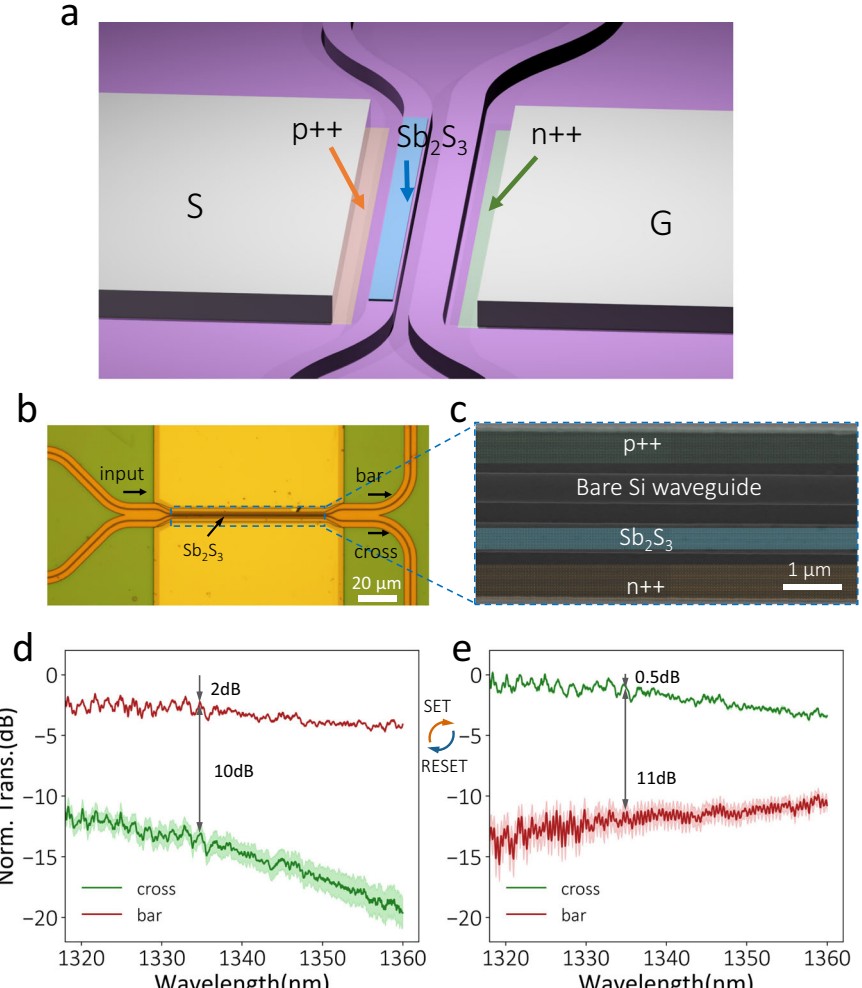

**Fig. 3 | An asymmetric directional coupler with Sb₂S₃-Si hybrid waveguide.**
**a** Schematic (the encapsulation ALD Al₂O₃ layer is not shown), **b** optical, and **c** SEM images of the asymmetric directional coupler. **d**, **e** Transmission spectra at bar and cross ports for (**d**) a-Sb₂S₃ (RESET: three 9.6 V, 500 ns, 922 nJ pulses) and (**e**) c-Sb₂S₃ (SET: three 2.7 V, 200 ms, 29.2 mJ pulses). The result was averaged over five measurements, and the shaded region indicates the standard deviation.

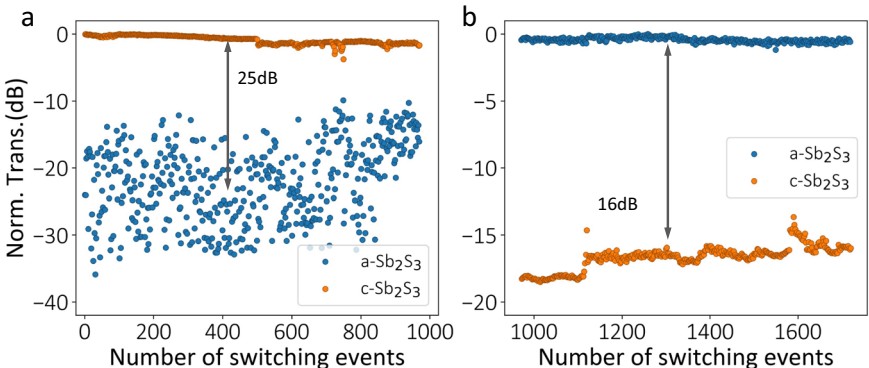

**Fig. 4 | Cyclability test for a-Sb₂S₃-based asymmetric directional coupler.**
Measured transmission at the (**a**) cross and (**b**) bar ports. The blue and orange scatterers represent the normalized transmission when Sb₂S₃ is in the amorphous and crystalline phases. The phase change condition is the same as in Fig.3. The device was switched for over 1600 events with no significant insertion loss and performance degradation.

We also experimentally tested the partial crystallization of the device and multilevel operation (Supplementary Fig. 10), which is based on the growth-dominant nature of Sb₂S₃ crystallization process[36]. In the following experiments, we mainly focused on partial amorphization because of lower energy consumption and more operation levels with finer resolution.

Such stepwise multilevel operation by applying identical pulses is distinctly different from previously reported multilevel operations in GST[22,24] and Sb₂Se₃[18,23], where different voltage amplitudes or pulse duration were used to access multiple levels during amorphization. The pulse-number-dependent behavior is quite counterintuitive: one expects that after the first amorphization pulse, the thin Sb₂S₃ film

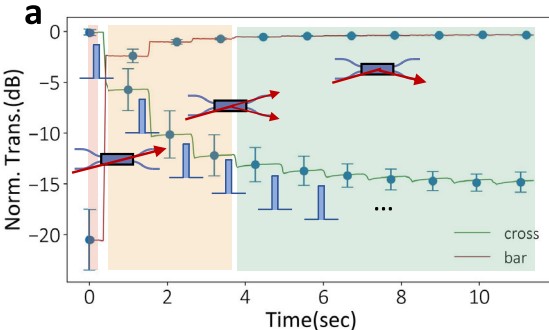
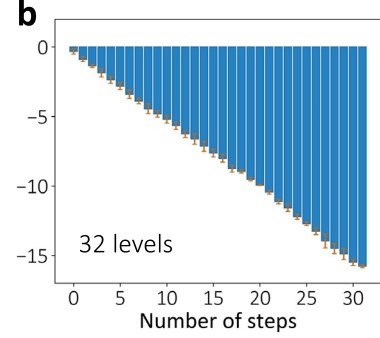

**Fig. 5 | A quasi-continuously tunable directional coupler based on multilevel Sb₂S₃. a** (Coarse tuning applying identical electrical pulses) Time trace measurement of the directional coupler when sending in an amorphization pulse (10 V, 550 ns, 1.1 µJ) each second. Green (red) curves represent cross (bar) transmission. The result is averaged over five experiments for each port, and the error bar shows the standard deviation. Depending on the pulses, a "bar", "intermediate", and "cross" state can be achieved as indicated by the red, orange, and green regions. **b** (Fine tuning using dynamically controlled near-identical electrical pulses)

Normalized transmission at 1,340 nm at the bar port shows 5-bit operation (32 distinct levels), achieved by dynamically controlling the number, amplitude, and duration of pulses sent in (near-identical, 9.65 V - 9.85 V, 550 ns, 1.02 µJ to 1.06 µJ). A precise transmission level of 0.50 ± 0.16 dB per step and 32 levels were simultaneously achieved. The only slight difference between the target and achieved transmission demonstrates an on-demand operation. The error bars represent the standard deviation over five experiments.

would have reached its new equilibrium phase. Moreover, since the thermal processes relax within 10 µs (Supplement Fig. 11), each pulse is independent due to the relatively long one-second interval. As a result, the subsequent identical and separated pulses should not further change the material phase. To understand the origin of the multi-level operation, we closely inspected four partially amorphized Sb₂S₃ devices under the microscope. We observed a few separate patches (Supplement Fig. 12) and a region that grew with more voltage pulses. As reported in some literature, one possibility could have been that a- and c-Sb₂S₃ have significantly different thermal conductivities and specific heat capacities. But we measured the thermal conductivities to be similar (a-Sb₂S₃: 0.2 W/m/K; c-Sb₂S₃: 0.4 W/m/K, see Methods), and hence, we ruled out this as a possible explanation. We hypothesize that this behavior comes from Sb₂S₃'s multiple crystalline phases. Sb₂S₃ has at least two distinct crystalline phases[39], which may differ in the amorphization conditions. The partial amorphization pulse can cause amorphization in the hottest region, but at the lower temperature region, it may cause phase transition to the other crystalline phase. These regions get amorphized in subsequent pulses, resulting in multilevel operation.

An even finer multilevel operation was realized by monitoring the transmission level and dynamically changing the pulse conditions slightly. Here, we demonstrate on-demand 5-bit operation in a quasi-continuously tunable directional coupler, as shown in Fig. 5b. We dynamically controlled the partial amorphization pulses to have slightly lower, near-identical voltages (ranging from 9.65 V to 9.85 V) and obtained up to 32 levels. Figure 5b demonstrates 5-bit operation (32 distinct levels) with a target resolution of 0.5 dB per level step at 1340 nm (see the detailed pulse conditions in Supplementary Table 2). We emphasize that dynamic control is necessary to mitigate the stochastic nature of electrically controlled PCMs[38], hence essential for a reliable many-level operation. Furthermore, the stepwise multi-level operation allows precise control by gradually approaching the desired operation level, whereas the voltage or pulse duration dependent approaches result in unrepeatable multi-level operation, limited by PCM's stochastic nature. In Fig. 5b, a linear fit shows a slope of −0.50 dB per step and a standard deviation of 0.16 dB among five experiments, indicating a repeatable operation. While the 5-bit operation of GST was shown using laser pulses[40], our demonstration is thus far the highest number of operating levels reported using electrical control in PCMs-based photonics. Moreover, our multilevel operation does not require sophisticated heater geometry engineering, such as the segmented doped silicon heater design[18], and solely relies on the phase-change

dynamics of Sb₂S₃. We note that the deviation from the ideal level mostly comes from the over-amorphization of the device, and the error can be reduced by incorporating a "backward" tuning using partial crystallization pulses. Implementing a more sophisticated heater geometry[18,41] could also potentially increase the number of levels.

### Random phase error correction in balanced MZIs exploiting multilevel operation

Finally, exploiting the multilevel operation, we corrected random phase errors in a balanced MZI. A perfectly balanced MZI should initially be in an all-cross state. However, random phase errors due to fabrication imperfections can easily build up to a phase error of π, making the initial state unpredictable. Therefore, balanced MZIs usually require extra calibration[35]. For example, Fig. 6a shows the transmission spectra of a phase error corrupted balanced MZI, indicating a high bar transmission. Both arms of the MZI are loaded with 40-µm-long Sb₂S₃ film to guarantee a phase tuning range of more than ±π. The corrected MZI spectra by multilevel tuning of Sb₂S₃ are demonstrated in Fig. 6b, showing a pure "cross state" with a high extinction ratio of 24 dB. The trimming process is shown in Fig. 6c. We sent in a partial amorphization pulse (8.8 V, 150 ns, 232 nJ) every other second, which gradually increased the portion of amorphous Sb₂S₃, resulting in quasi-continuous changes of the bar (blue) and cross (orange) transmission (at 1340 nm). The correction finishes once a bar transmission minimum is reached, indicated by the red arrows in Fig. 6c, and the spectra reported in Fig. 6b were then measured. Further pulses increase bar transmission because of the over-compensated phase. We performed the same experiment three times, indicated by different colored regions in Fig. 6c. Complete phase error correction was observed in all three instances, exhibiting excellent repeatability of our trimming process. Note that binary tuning cannot accomplish this task due to the random initial phase error. Even multilevel operations with limited discrete-level resolution can cause over- or under-corrected phase error, ultimately determining the trimming resolution. We highlight that this method requires zero static energy supply once the phase error is corrected, as the phase transition in Sb₂S₃ is non-volatile (>77 days, Supplementary Fig. 13). Thanks to the relatively fine operation levels, slight over-tuning does not significantly affect the performance. The trimming resolution can be further improved using dynamically changed, near-identical electrical pulses, as shown in the previous Section. Moreover, if the phase error is over compensated, the device could be tuned back with

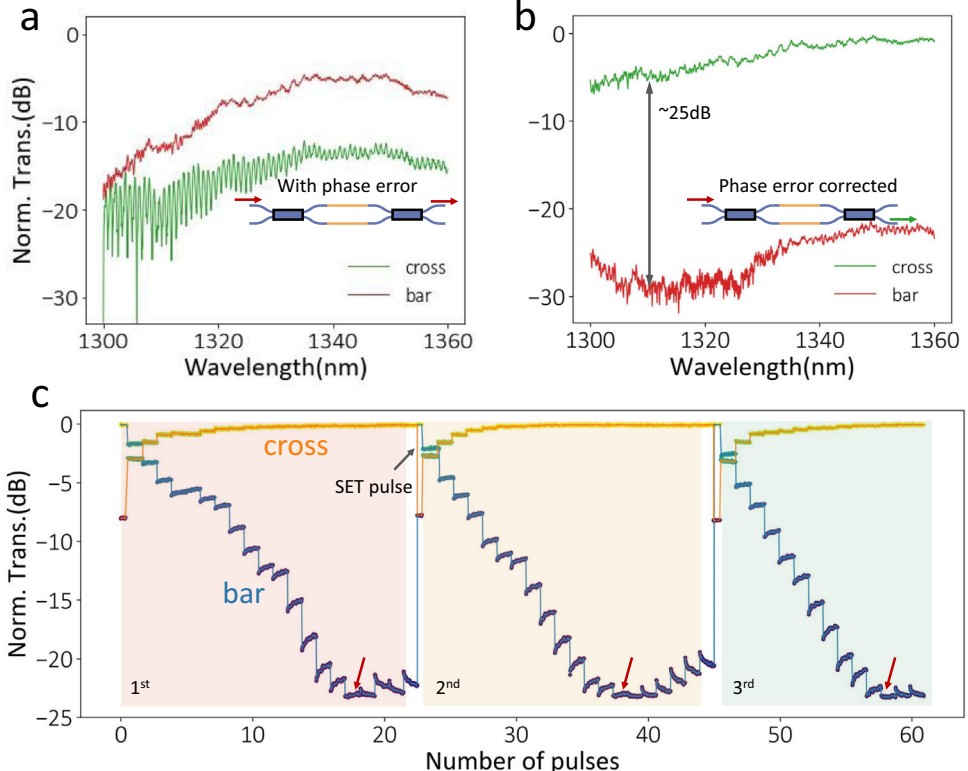

**Fig. 6 | Random phase error correction in a balanced MZI based on the multi-level operation. a** Transmission spectrum of a balanced MZI with phase error due to fabrication imperfections. The red (green) line represents bar (cross) transmission. Light transmits mostly from the bar port instead of the ideal cross port. **b** Transmission spectrum after the correction with 8.8 V, 150 ns, 232 nJ pulses. The device is in a "cross state" with a high extinction ratio of -24 dB, indicating a phase-error-free device. **c** Time trace measurement for three experiments shows the phase error correction step. The RESET pulses are sent at a one-second interval. As

more pulses were sent in, the bar transmission continuously decreased until it reached a minimum of -–24 dB (red arrow), and then it went up. In all the experiments, the optimal error correction was obtained. The slight volatile increase (only visible when transmission is <−12 dB) after each amorphization pulse happens over a much longer timescale than 10 μs thermal relaxation time and could be attributed to a weak persistent photocurrent effect and requires more study. However, the transmission stabilizes before the next pulse. The blue (orange) represents bar- (cross-) transmission.

partial crystallization pulses. In the future, our trimming process can potentially be fully automated by real-time adjusting the pulse numbers according to the measured transmission.

## Discussion

Among the first few cycles (-50 cycles), sub-millisecond pulses failed to trigger the crystallization for $Sb_2S_3$. We increased the voltage of a 500-μs pulse up to the material ablation voltage level without observing any crystallization. This suggests that the thermally induced $Sb_2S_3$ crystallization process is relatively slow, different from laser-induced crystallization[28], where a crystallization speed of tens of nano-second was demonstrated. We attribute this behavior to the difficulty in the initial nucleation. This hypothesis is supported by the fact that after the first few crystallization processes (conditioning), consecutive crystallization process could happen only at 100 μs (Supplementary Fig. 15), increasing the speed and lowering the energy by three orders of magnitudes. Hence, crystallization becomes easier after the initial nuclei/ defects are formed by the first few long thermal processes. We used 200-ms pulses in the reported experiments because the post-conditioning, shorter 100-μs pulse condition was identified later in our experiments.

However, this limited crystallization speed (-kHz) should not be considered as a problem, because of PCMs' main advantage of non-volatility, and thus zero-static power consumption. This nonvolatile reconfiguration technology is complementary to the traditional high-speed modulators, and will find wide applications with infrequent programming needs, where high speed is not essential and the zero-static power is more important[10,42]. Moreover, the slow crystallization

speed could prevent unintentional recrystallization during long-duration amorphization, which has a slower cooling rate. Since long-duration amorphization means a smaller vertical thermal gradient, a thicker layer of $Sb_2S_3$ can potentially be switched completely. We verified that a pulse with 10 μs duration was able to trigger a large degree of amorphization (Supplementary Fig. 16), which is not possible in fast PCMs, such as GST. This capability of actuating phase transition in thicker films is useful in photonics, since switching a larger volume of PCMs is generally required to provide stronger modulation. In addition, such long amorphization pulse reduces the voltage from >10 V (for 500 ns pulses) to -5 V, showing a critical step toward CMOS-compatible operation levels (1 - 2 V). The voltage could be further reduced by designing the heater geometry to improve the thermal delivery efficiency[43], such as etching the oxide beneath the silicon[44], adding thermal insulation[45] layers or using atomically thin 2D material heaters[23]. Lastly, we compare our device with other PCM devices and other tuning methods in Supplementary Tables 4, 5.

In conclusion, we demonstrated a multi-bit integrated electro-photonic platform using wide bandgap PCM $Sb_2S_3$ and doped silicon PIN heaters. Operating in the telecommunication O-band, the $Sb_2S_3$-Si hybrid ring resonators, MZIs, and directional couplers exhibit a low insertion loss (<1.0 dB) and a high extinction ratio (> 10 dB). We report record-high electrical cyclability of $Sb_2S_3$ (>1600 switching events). Notably, the $Sb_2S_3$-based photonic devices could be tuned into distinct intermediate levels in a stepwise fashion by consecutively sending identical, thermally isolated partial amorphization pulses. Such pulse-number-dependent multilevel tuning allows high-resolution operation levels compared to different voltages or pulse widths. We demonstrate

precise on-demand 5-bit operation with 0.50 ± 0.16 dB per level step in a tunable beam splitter. We further exploited this behavior to demonstrate random phase error correction in a balanced MZI. Our work shows promise for various integrated photonics applications, such as post-fabrication trimming, optical information processing, and optical quantum simulations, where the nonvolatile on-demand multi-bit operation is of paramount interest.

## Methods

### SOI device fabrication

The silicon photonic devices were fabricated on a commercial SOI Wafer with 220-nm-thick Silicon on 2-μm-thick $SiO_2$ (WaferPro). All devices were defined using electron-beam lithography (EBL, JEOL JBX-6300FS) with a positive-tone E-beam resist (ZEP-520A) and partially etched by ~120 nm in a fluorine-based inductively coupled plasma etcher (ICP, Oxford PlasmaLab 100 ICP-18) with mixed $SF_6$/$C_4F_8$. The etching rate was around 2.8 nm/sec. The doping regions were defined by two additional EBL rounds with 600-nm-thick poly (methyl methacrylate) (PMMA) resist and implanted by boron (phosphorus) ions for $p^{++}$ ($n^{++}$) doping regions with a dosage of $2 \times 10^{15}$ ions per $cm^2$ and ion energy of 14 keV (40 keV). The chips were annealed at 950 °C for 10 min (Expertech CRT200 Anneal Furnace) for dopant activation. Ideal Ohmic contact was formed after removal of the surface native oxide via immersing the chips in 10:1 buffered oxide etchant (BOE) for 10 s. The metal contacts were then immediately patterned by a fourth EBL step using PMMA. Metallization was done by electron-beam evaporation (CHA SEC-600) and lift-off of Ti/Pd (5 nm/180 nm). We measured a 30-μm-long PIN diode and obtained a threshold voltage of around 0.8 V, and a resistivity of around 50 Ω. After a fifth EBL defining the $Sb_2S_3$ window, a 40-nm $Sb_2S_3$ thin film was deposited using a GST target (AJA International) in a magnetron sputtering system (Lesker Lab 18), followed by a lift-off process. We note the actual $Sb_2S_3$ thickness on the waveguide was reduced to around 20 nm because of the narrow PMMA trench (Supplementary Table 3). The $Sb_2S_3$ was then encapsulated by 40-nm-thick $Al_2O_3$ through thermal ALD (Oxford Plasmalab 80PLUS OpAL ALD) at 150 °C. To ensure good contact between the electric probe and metal pads while applying electrical pulses, the $Al_2O_3$ on the metal contacts was removed by defining a window using a sixth EBL with 600 nm PMMA, then etching in a chlorine-based inductively coupled plasma etcher (ICP-RIE, Oxford PlasmaLab 100 ICP-18).

### Optical simulation

The refractive index data for $Sb_2S_3$ were measured by an ellipsometer (Woollam M-2000)[20]. The phase shifters and asymmetric directional couplers were designed (verified) by a commercial photonic simulation software package Lumerical MODE (FDTD).

### Heat transfer simulation

The doped silicon PIN heater was simulated with a commercial Multiphysics simulation software COMSOL Multiphysics[24,46]. In the simulation, a heat transfer in a solid model is coupled with a semiconductor model to simulate the transient time performance.

### Optical transmission measurement setup

The programmable units were measured with a 25°-angled vertical fiber-coupling setup. The stage temperature was controlled at 26 °C by a thermoelectric controller (TEC, TE Technology TC-720). A tunable continuous-wave laser (Santec TSL-510) sent in the input light, the polarization of which was controlled by a manual fiber polarization controller (Thorlabs FPC526) to achieve a maximum fiber-to-chip coupling efficiency. A low-noise power meter (Keysight 81634B) measured the static optical transmission. The transmission spectra of all $Sb_2S_3$ devices were normalized to the spectra of the nearest reference waveguide. For the on-chip electrical switching, electrical pulses were applied to the on-chip metal contacts via a pair of electrical probes on two probe positioners (Cascade Microtech DPP105-M-AI-S). The crystallization and amorphization pulses were generated from a pulse function arbitrary generator (Keysight 81160 A). The tunable laser, power meter, thermal controller, source meter, and pulse function arbitrary generator were controlled by a LabView program[24].

When electrically switching $Sb_2S_3$, we found that a single amorphization or crystallization voltage pulse switches the device to transmission levels with large state-to-state variations (Supplementary Fig. 14). This could be attributed to the switching of $Sb_2S_3$ into random intermediate structural phases since it has two (or more) crystalline phases[39]. This issue was tackled by using three identical pulses to switch the $Sb_2S_3$ phase completely.

### Thermal parameter measurement

The thermal conductivity of our films was measured with time-domain thermoreflectance (TDTR)[47,48]. TDTR uses ultrafast modulated laser heating through the absorption of a thin metallic transducer layer (70 nm Pt). An unmodulated probe laser then measures the surface temperature through a proportional change in the transducer reflectivity. Measurements were taken at a pump beam modulation frequency of 4 MHz and 10 MHz to ensure the thermal penetration depth exceeded the thickness of the films. Knife edge measurements provided $1/e^2$ beam radii of 5.5 and $3.1 \pm 0.05$ μm for the pump and probe, respectively. The resulting thermoreflectance data were then fit to the solution of a 3D heat diffusion model for a multi-layer stack of materials (Pt-film-substrate) and the effective thermal resistance was determined as a function of film thickness. A linear regression of the foregoing data provided the thermal boundary resistance of the material stack, which was then used to determine the intrinsic thermal conductivities for the films (Supplementary Figs. 17–19). The properties of the Pt layer, the films, and the Si substrate were determined from independent measurements or adopted from the literature[49,50].

## Data availability

The data that support the findings of this study are available from the corresponding author upon request.

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

## Acknowledgements

The authors thank Asir Intisar Khan, Kathryn M. Neilson, Prof. Eric Pop at Stanford University, and Prof. Juejun Hu at Massachusetts Institute of Technology for the insightful discussions. The research is funded by National Science Foundation (NSF-1640986, NSF-2003509), ONR-YIP Award, DARPA-YFA Award, NASA-STTR Award 80NSSC22PA980, and Intel. F.M. is supported by a Draper Scholars Program. Part of this work was conducted at the Washington Nanofabrication Facility/Molecular Analysis Facility, a National Nanotechnology Coordinated Infrastructure (NNCI) site at the University of Washington, with partial support from the National Science Foundation via awards NNCI-1542101 and NNCI-2025489. Part of this work was performed at the Stanford Nano Shared Facilities (SNSF), supported by the National Science Foundation under award ECCS-1542152).

## Author contributions

R.C. and A.M. conceived the project. R.C. simulated and fabricated the Sb2S3 silicon photonic devices, performed optical characterizations

and data analysis. Z.F. and J.Z. helped with the device simulation, fabrication, characterization, and data analysis. F.M., S.J.G., K.K. and A.S. helped with the optical measurements. C.P. and K.E.G. measured thermal conductivity of Sb2S3 thin films. A.M. supervised and planned the project. R.C. wrote the manuscript with input from all the authors.

## Competing interests

The authors declare no competing interests.
