## [Peer Review File · Nature Communications]

Non-volatile electrically programmable integrated photonics with a 5-bit operationEditorial Note: This manuscript has been previously reviewed at another journal that is not operating a transparent peer review scheme. This document only contains reviewer comments and rebuttal letters for versions considered at *Nature Communications*.

Reviewer #1 (Remarks to the Author):

This a resubmission of a paper previously submitted to Nature Photonics. The authors have responded to the reviewers comments.

I have read all reviewer questions and the responses and I have to admit that the authors have responded thoughtfully. They have further made correction to the manuscript and improved it significantly.

I have an optional comment though. Their response to reviewer 1 comment "Yet, the material system is not novel. Sb₂S₃ has been explored before." is well worded. The authors differentiate their own work and compare it against published work. This passage in the reviewer's response is better over what is found in the paper. It might have been a good idea to bring in some of these comments into the revised version as well.

In conclusion, I think that this paper is now in a good shape and deserves to be published.

Reviewer #2 (Remarks to the Author):

I am satisfied with the changes the authors have made and recommend publication in Nature Communications.

(I am also looking forward to the authors' upcoming work with Intel!)

Response letter to the reviewers of the article “**Non-volatile electrically programmable integrated photonics with a 5-bit operation**”

First, we would like to thank the reviewers for their time and effort to carefully reading our paper and providing us with constructive criticisms. Their comments have surely helped us improve the manuscript. We appreciate the reviewers for commenting that we “*have responded thoughtfully*”, “*made correction to the manuscript and improved it significantly*” and glad to see that reviewer 2 is “*satisfied with the changes*”. Despite these compliments, we have also taken their criticisms seriously and have modified our manuscript accordingly to take account of all suggestions from the reviewers. Below you can find our point-by-point replies to all the reviewers’ remarks. The changes are **highlighted** in the revised manuscript.

Reviewer 1:

This a resubmission of a paper previously submitted to Nature Photonics. The authors have responded to the reviewers comments. I have read all reviewer questions and the responses and I have to admit that the authors have responded thoughtfully. They have further made correction to the manuscript and improved it significantly.

Response: We thank the reviewer for commenting that we “*have responded thoughtfully*”, “*made correction to the manuscript and improved it significantly*”. We have responded to the comments below.

I have an optional comment though. Their response to reviewer 1 comment "Yet, the material system is not novel. Sb₂S₃ has been explored before." is well worded. The authors differentiate their own work and compare it against published work. This passage in the reviewer's response is better over what is found in the paper. It might have been a good idea to bring in some of these comments into the revised version as well.

Response: We thank the reviewer for the advice. We have added some comments to the Introduction section.

Actions taken: We have added the following sentences to the Introduction section.

Despite these promises, previous works mostly switched a Sb_2S_3 blanket film optically^{29,34} or a Sb_2S_3 metasurface^{31,32}, usually with sophisticated femto-second pulse lasers. Very few electrical controls of Sb_2S_3 were experimentally demonstrated^{26,29} and high-endurance electrical control of Sb_2S_3 remained unsolved.

In conclusion, I think that this paper is now in a good shape and deserves to be published.

Response: We thank the reviewer for finding the paper is “*in a good shape and deserves to be published*”. We want to gratefully appreciate the suggestions and comments the reviewer provided along the way, which has improved the paper significantly.

Reviewer 2:

I am satisfied with the changes the authors have made and recommend publication in Nature Communications. (I am also looking forward to the authors' upcoming work with Intel!)

Response: We are glad that the reviewer is “*satisfied with the changes the authors have made and recommend publication in Nature Communications*”. We are also working hard with Intel and hope to show interesting systems using electrically controlled Sb_2S_3 . We want to gratefully appreciate the suggestions and comments the reviewer provided along the way, which has significantly improved the paper significantly.